# Prevalence and Antifungal Susceptibility of Clinically Relevant *Candida* Species, Identification of *Candida auris* and *Kodamaea ohmeri* in Bangladesh

**DOI:** 10.3390/tropicalmed7090211

**Published:** 2022-08-26

**Authors:** Fardousi Akter Sathi, Shyamal Kumar Paul, Salma Ahmed, Mohammad Monirul Alam, Syeda Anjuman Nasreen, Nazia Haque, Arup Islam, Sultana Shabnam Nila, Sultana Zahura Afrin, Meiji Soe Aung, Nobumichi Kobayashi

**Affiliations:** 1Department of Microbiology, Mymensingh Medical College, Mymensingh 2200, Bangladesh; 2Netrokona Medical College, Netrokona 2400, Bangladesh; 3Department of Microbiology, Mugda Medical College, Dhaka 1214, Bangladesh; 4Department of ENT, Mymensingh Medical College Hospital, Mymensingh 2200, Bangladesh; 5Department of Hygiene, School of Medicine, Sapporo Medical University, Sapporo 060-8556, Japan

**Keywords:** *Candida*, *C. albicans*, *C. auris*, *Kodamaea ohmeri*, fluconazole resistance, *ERG11*, Bangladesh

## Abstract

*Candida* species are major fungal pathogens in humans. The aim of this study was to determine the prevalence of individual *Candida* species and their susceptibility to antifungal drugs among clinical isolates in a tertiary care hospital in Bangladesh. During a 10-month period in 2021, high vaginal swabs (HVSs), blood, and aural swabs were collected from 360 patients. From these specimens, *Candida* spp. was isolated from cultures on Sabouraud dextrose agar media, and phenotypic and genetic analyses were performed. A total of 109 isolates were recovered, and *C. albicans* accounted for 37%, being derived mostly from HVSs. Among non-albicans *Candida* (NAC), *C. parapsilosis* was the most frequent, followed by *C. ciferrii*, *C. tropicalis*, and *C. glabrata*. Three isolates from blood and two isolates from aural discharge were genetically identified as *C. auris* and *Kodamaea ohmeri*, respectively. NAC isolates were more resistant to fluconazole (overall rate, 29%) than *C. albicans* (10%). *Candida* isolates from blood showed 95% susceptibility to voriconazole and less susceptibility to fluconazole (67%). Two or three amino acid substitutions were detected in the *ERG11* of two fluconazole-resistant *C. albicans* isolates. The present study is the first to reveal the prevalence of *Candida* species and their antifungal susceptibility in Bangladesh.

## 1. Introduction

*Candida* spp. is ubiquitous yeast and exists as normal flora within the mouth, throat, intestine, genital, and urinary tracts of humans. However, they can cause a broad spectrum of human infections, known as candidiasis, which include superficial (oral thrush, vulvovaginal candidiasis, otomycosis, paronychia, etc.) and deep-seated fungal infections (e.g., candidemia) [1]. Though the genus *Candida* includes more than 200 species, only 15 species have been isolated from infections in humans and animals [2]. *Candida albicans* has generally been the dominant species, accounting for about one half or more of the isolates from invasive infections, including candidemia, vulvovaginal candidiasis, denture stomatitis, and so forth [3,4,5]. Among non-albicans *Candida* (NAC) species, *C. glabrata*, *C. parapsilosis*, *C. tropicalis*, and *C. krusei* are the most frequently identified among clinical isolates, and their increasing trend has been noted [3,4,6,7,8].

During the past several decades, the incidence of candidiasis has substantially increased; this increase has been associated with the progress of medical care that includes aggressive antibiotic therapy practices and the use of immunosuppressive agents [9,10]. For example, treatment with interleukin 17 inhibitors for psoriatic patients increases the risk of developing fungal infections [11]. *C**andida* spp. is the fourth most common cause of nosocomial bloodstream infection in the US [12], and candidemia causes a high mortality rate (28–38%) [13,14,15]. Recently, an increase in the isolation of NAC, associated with a relative decrease in *C. albicans*, has been observed [7,10,13,16]. Accordingly, the rising trend of resistance to antifungals is a concern for NAC, especially *C. glabrata* and *C. parapsilosis* [7,16,17]. Notably, *C. auris* has occurred and spread globally as an emerging pathogen that shows multidrug resistance and causes invasive infections in nosocomial settings [18].

Almost all *Candida* spp. are prevalent globally, with *C. albicans* being dominant and accounting for 30–70% of all *Candida* spp. from candidemia and invasive candidiasis. Nevertheless, particular geographical distributions and patient type-specificity have been observed for individual NAC species [19,20]. *C. parapsilosis* is primarily reported in Australia, Latin America, and Mediterranean countries, isolated from neonates and young adults, and often associated with the presence of a central venous catheter. *C. glabrata* is dominantly distributed in the US and north and central Europe, while *C. tropicalis* is dominantly found in Asia, the Middle East, and a part of Latin America. *C. krusei* is relatively more prevalent in Brazil, Canada, some European countries, and Australia. *C. glabrata*, *C. tropicalis*, and *C. krusei* are usually derived from older patients and are associated with solid tumors, organ transplants, and abdominal surgery.

For the treatment of *Candida* infections, several classes of antifungals, including azoles, polyenes, and echinocandins, are available. Among the antifungals, fluconazole, a member of the azole class compound, has most commonly been used. Azoles inhibit the fungal lanosterol 14 α-demethylase (*ERG11*), leading to a reduction in the production of ergosterol, an essential fungal membrane sterol. *Candida* species acquire resistance to azoles through the development of mutations in this enzyme, which reduces the binding affinity to azoles [21,22].

In Bangladesh, little is known about the actual situation of infections caused by *Candida* species, though the considerable burden of candidemia has been estimated [23]. The present study was conducted to clarify the prevalence of individual *Candida* species in different types of infections and their susceptibility to antifungal drugs among clinical isolates in a tertiary care hospital. Information regarding the prevalence of *Candida* species, based on accurate identification within this study, may provide advancement in the overview of candidiasis in Bangladesh for a better understanding of its etiology and antifungal treatment. Particularly, the novel findings, including the identification of emerging *Candida* species and *ERG11* mutations, may inform clinicians of public health issues related to *Candida* in the country.

## 2. Materials and Methods

### 2.1. Study Design and Setting, Collection of Specimens

This research was conducted as a cross-sectional, observational study in a single center within Mymensingh Medical College hospital. Three types of clinical specimens (i.e., high vaginal swabs (HVSs), blood, and aural swabs (discharge)) were collected from patients with suspected vulvovaginal candidiasis (inpatients and outpatients; department of gynecology and obstetrics), candidemia (inpatient; department of pediatrics, Intensive Care Unit (ICU) and Neonatal Intensive Care Unit (NICU), hematology, oncology, medicine, and surgery), and otomycosis (outpatients; ear, nose, and throat (ENT) department), respectively, from March to December 2021. The inclusion criteria of patients with each infection type were as follows: patients clinically suspected of candidemia, showing symptoms of sepsis not responding to antibiotic treatment for 7 days, and possessed previously described risk factors for candidemia [24,25]; women within the age group of 18–56 years, with clinically suspected vulvovaginal candidiasis, and possessing risk factors (diabetes mellitus, pregnancy, use of oral contraceptive, or antibiotics); both male and female patients of all age groups with clinically suspected *Candida*-associated otomycosis.

### 2.2. Culture and Identification of Candida spp.

Blood cultures were taken for all patients with suspected candidemia (i.e., patients with clinical signs or symptoms of sepsis with specific risk factors for candidemia, as mentioned above). Blood samples were inoculated aseptically into BACT/ALERT^®^FA/PF PLUS blood culture bottles (bioMérieux, Durham, NC, USA) and incubated in a BACT/ALERT machine at 37 °C until a positive indication of growth was provided by the machine or for a maximum of 5 days [26]. Subsequently, liquid media containing blood from the BACT/ALERT^®^FA/PF PLUS bottles were cultured on Sabouraud dextrose agar (SDA) media with chloramphenicol (Sabouraud chloramphenicol agar, HiMedia, Mumbai, India); blood agar media and MacConkey agar media were housed at 37 °C for 24 h and up to 48 h for SDA media. Chloramphenicol was supplemented with SDA media to inhibit bacterial growth. HVSs and aural discharges were cultured on SDA media with chloramphenicol at 35–37 °C for 24–48 h, aerobically.

All the *Candida* isolates were identified based on characteristics of a colony (creamy white, smooth, pasty consistency) on SDA media with a distinct smell. All suspected colonies were examined using standard microbiological methods, including microscopic examination, Gram staining, germ tube tests, subcultures on chromogenic agar media (HiCrome™Candida Differential Agar M1297A, HiMedia, Mumbai, India), citrate utilization tests, and urea hydrolysis tests. For all of the isolates, species were identified genetically through PCR detection of the ITS1-5.8S-ITS2 ribosomal region with ITS1 and ITS4 primers, and its RFLP profile after digestion with MspI, as described previously [27]. For isolates left unidentified after using the PCR-RFLP method, the nucleotide sequence of the PCR product was determined using the Sanger sequencing method, and the identical or most similar sequence was searched for using the BLAST web tool (https://blast.ncbi.nlm.nih.gov/Blast.cgi, accessed on 20 February 2022). The assigned species was confirmed phylogenetically, with sequences of other strains retrieved from the GenBank database through the use of MEGA.11 software.

### 2.3. Antifungal Susceptibility Testing

Susceptibility to antifungal agents was determined using the broth microdilution method and the disk diffusion method. The minimum inhibitory concentration (MIC) to fluconazole was determined for all isolates, while MIC to amphotericin B was only measured for blood isolates. Based on the disk diffusion method using commercial disks (HiMedia, Mumbai, India), susceptibility to fluconazole (25 μg), voriconazole (1 μg), itraconazole (10 μg), and amphotericin B (10 μg) was tested for among blood isolates. In contrast, susceptibilities to fluconazole (25 μg), itraconazole (10 μg), clotrimazole (10 μg), and nystatin (100 U) were examined for among isolates from HVSs and aural swabs. Susceptibilities to fluconazole and voriconazole were interpreted using the CLSI standard M60-Ed2 [28], and those of other antifungals were judged according to the published criteria [29,30]. For NAC species for which the standards of fluconazole susceptibility are not defined by the CLSI, the criterion of *C. albicans* was applied as described previously [31].

### 2.4. Sequence Analysis of the ERG11 Gene

As an additional analysis, a nucleotide sequence of the *ERG11* gene was determined for *C. albicans* isolates showing fluconazole resistance using the PCR and Sanger sequencing methods. For PCR, previously reported primers were used [32,33]. Sequence data were compared with those of fluconazole-susceptible strains through alignment using the Clustal Omega program (https://www.ebi.ac.uk/Tools/msa/clustalo/, accessed on 10 April 2022).

### 2.5. Statistical Analysis

Patients’ information was collected through clinical records and structured questionnaires. The risk factors of each infection type and the data obtained in the present study were statistically analyzed using IBM SPSS Statistics for Windows Version 26 (IBM Corp. Armonk, NY, USA).

### 2.6. GenBank Accession Numbers

The *ERG11* gene sequences of *C. albicans* determined in this study were deposited to GenBank under accession numbers ON161125 and ON161126.

## 3. Results

During the 10-month study period, 360 samples of suspected patients with candidiasis (HVSs = 175; blood = 125; aural swabs = 60) were examined, and a total of 109 isolates of *Candida* were recovered from HVSs (n = 52), blood (n = 39), and aural swabs (n = 18). Vulvovaginal candidiasis was diagnosed mainly within the reproductive age group (18–45 years) and candidemia patients were mostly neonates, while otomycosis was found within a wide age range (Appendix A). For 104 isolates, 11 species of *Candida* were identified phenotypically as well as through PCR and RFLP methods (Appendix A). The remaining five isolates were identified through a sequence analysis of the ITS1-5.8S-ITS2 region. After all, 12 *Candida* species and *Kodamaea ohmerii* were identified (Table 1).

*C. albicans* (n = 40) accounted for 37% of all isolates and was mostly derived from HVSs (34/40, 85%) as well as major species from HVSs (34/52, 65.4%) (Table 1). Among NAC (n = 69), *C. parapsilosis* was the most frequent (n = 25, 36%), followed by *C. ciferii*, *C. tropicalis*, and *C. glabrata*. The dominant species in blood isolates was *C. parapsilosis* (n = 16, 41%), with *C. ciferrii* (n = 9, 23%) being the second most common. Half of the isolates from the aural swabs were *C. parapsilosis* or *C. tropicalis*. Three isolates from blood and two isolates from aural swabs were identified as *C. auris* and *K. ohmeri*, respectively. The ITS1-5.8S-ITS2 region of these isolates clustered with those of individual species in phylogenetic analysis, showing >99% identity (Appendix A). *K. ohmeri* was derived from otomycosis in female patients of 40 and 42 years of age. All the *C. auris* were isolated from neonates, including two fatal cases. The first fatal case with a *C. auris* infection was a 15-day-old male neonate with signs of sepsis not responding to antibiotics (meropenem and gentamicin for 10 days; later on, colistin); the neonate died 2 days after the diagnosis of multiorgan failure due to sepsis, though voriconazole was used in treatment. The isolated *C. auris* was resistant to fluconazole, itraconazole, voriconazole, clotrimazole, nystatin, and amphotericin B. The second fatal case was a 2-day-old male neonate with signs of early-onset neonatal sepsis—who was treated with ceftazidime, amikacin, and, later, voriconazole—but died from cardio-respiratory failure due to septicemia. A survival case was a 12-day-old male neonate, who showed symptoms of sepsis for 10 days without responding to ceftazidime and amikacin; he was later treated with colistin with no response. He recovered 7 days after treatment with voriconazole.

Table 2 shows susceptibility to fluconazole among all of the *Candida* isolates, based on MIC (the distribution of the MIC values is shown in Appendix A). Although most *C. albicans* isolates were susceptible or susceptible dose dependent (SDD), NAC exhibited significantly higher resistance rates (overall, 29%) than *C. albicans* (10%). Among NAC, fluconazole resistance was detected in eight species, among which *C. ciferrii*, *C. auris*, and *K. ohmeri* showed higher resistance rates (>67%) than other species, except for the intrinsically resistant *C. krusei*.

Antifungal susceptibility in each specimen (infection) type is shown in Table 3. The blood isolates exhibited 95% susceptibility to voriconazole, while showing lower susceptibility to fluconazole and itraconazole (67–69%). In contrast, isolates from HVSs and aural swabs were more susceptible to fluconazole as well as nystatin (78–94%), with lower susceptibility to clotrimazole. The distribution of the MICs of amphotericin B (blood isolates) is indicated in Appendix A.

The *ERG11* gene was analyzed for two *C. albicans* isolates (A173-22 from recurrent vulvovaginitis in a 27-year-old female patient; A1201-57 from candidemia in a male, premature, low-birth-weight neonate). These isolates were resistant to fluconazole, with a MIC of >64 μg/mL. Nearly a full-length sequence of the *ERG11* gene was determined for the *C. albicans* isolates and was compared with that of a wild type of the *ERG11* in the SC5314 strain, which is susceptible to fluconazole [34]. It was revealed that *C. albicans* isolates A173-22 and A1201-57 had three and two missense mutations, respectively, causing amino acid substitutions (A114V, F145L, and L276V in A173-22; A114V and F145L in A1201-57) (Appendix A).

Identified risk factors in patients with individual candidal infections are summarized in Table 4. In candidemia, the prolonged use of broad-spectrum antibiotics was found in all patients; subsequently, total parenteral nutrition and lower uterine cesarean section were also common among neonate cases. The oral contraceptive pill was most strongly associated with vulvovaginal candidiasis, while habitual cleaning and the use of antibiotic/steroid drops were frequently observed in otomycosis.

## 4. Discussion

In the present study, *Candida* species were identified among isolates from three infection types in a tertiary care hospital in Bangladesh, and their antifungal susceptibility was determined. Among isolates from HVSs, the predominant species was *C. albicans*, which has previously been reported as a cause of vulvovaginal candidiasis [35,36]. In contrast, from the aural swabs, *C. parapsilosis* was determined to be the most common, followed by *C. tropicalis*. Otomycosis is known to be caused by various fungal pathogens, with *Aspergillus* species being more common than *Candida* spp. [37]; thus, the prevalence of *Candida* species in otomycosis is not yet well understood. As for *Candida* spp. from otomycosis, a high prevalence of *C. parapsilosis* [37,38] as well as *C. albicans* [39] was described. The present study may support a major role of *C. parapsilosis* in otomycosis.

It was notable in the present study that *C. parapsilosis* and *C. ciferrii* were the species most frequently isolated from blood, accounting for >60% isolates of these species. Among *Candida* species, *C. albicans* is the main pathogen of bloodstream infections, as described in many studies [3,4,14]. However, an increase in NAC species has been observed recently [8], and *C. parapsilosis* was detected as the second most common species, showing a comparable proportion to *C. albicans* in some studies [40,41]. Furthermore, the highest detection rate was described for *C. parapsilosis* as a cause of candidemia [8,42]. In a study of candidemia in neonatal ICU patients in Spain, an increasing trend of *C. parapsilosis* was indicated over twenty years [42]. Because our study was conducted in a single hospital and most blood isolates were obtained from neonates, *C. parapsilosis* is suggested to have been persisting as a nosocomial pathogen, though the geographical differences in the prevalence of the *Candida* species may be related to its dominance.

*C. ciferrii* is a rare species of the human pathogen and causes various diseases, including systemic mycosis in immunocompromized hosts [43], pneumonia [44], endophthalmitis [45], and onychomycosis [46]. Although this species is not listed as a common cause of candidemia [3], there are a few case reports of fungemia due to *C. ciferrii* in children and adults [47,48]. In addition, *C. ciferrii* resulting from human infections has often shown fluconazole resistance [43,47]. Likewise, *C. ciferrii* isolates in the present study exhibited high resistance rates to fluconazole and itraconazole. The isolation of this species within three types of specimens in the present study may suggest the prevalence of this organism nosocomially or endemically in the study site, indicating the need for further surveillance.

*C. auris* and *K. ohmeri* are emerging fungal pathogens that cause nosocomial infections, lead to high mortality, and have been detected worldwide [18,49]. In Bangladesh, these species were rarely detected, with only *C. auris* being reported in a single study in Dhaka [50]. In the present study, *C. auris* and *K. ohmeri* were identified in candidemia in children and aural infections in adults, respectively. It has been revealed that *C. auris* is prone to causing invasive infections associated with resistance to multiple antifungal drugs, which is implicated in the difficulty of the treatment of its infection [18]. In accordance with these observations, all of the *C. auris* isolates in the present study were derived from bloodstream infections in neonates, including some fatal cases, and showed resistance to fluconazole, itraconazole, and amphotericin B; resistance to voriconazole was even shown in one isolate. Among the fatal cases, the recognition of non-response to the initial treatment with antibacterial drugs and the subsequent change to antifungals appeared to occur too late. Accordingly, early diagnosis and initiation of antifungal treatment may be essential for *C. auris* infections. *C. auris* is able to persist and survive on biotic and abiotic surfaces for long periods due to its unique traits, thermotolerance, and osmotolerance [18]. Accordingly, this *Candida* species is suggested to potentially remain viable within hospital environments and medical devices. To prevent the spread of *C. auris* in hospitals, early identification of this *Candida* species, enhanced personal protection practices, environmental cleaning, and the decolonization of patients may be required. Confirmation of *C. auris* in this study may be indicative of the necessity for preparation to control its infections within Bangladesh. *K. ohmeri* has been increasingly reported as a cause of invasive infections worldwide and is more frequently reported in Asia [49]. The identification of *K. ohmeri* in the present study within non-invasive infections may pose potential concerns for its distribution into the environment and the occurrence of invasive disease. Thus, the same control measures used for *C. auris* may be also necessary for *K. ohmeri*.

Although *C. albicans* from candidemia has been shown to be highly susceptible to fluconazole in worldwide scale surveillance (>99% susceptibility) [4,7,51], some regional studies (China and Turkey) have reported a 5–20% resistance rate to fluconazole [17,41,52]. In contrast, higher resistance rates to fluconazole (34–50%) were described among isolates from vulvovaginal candidiasis or various candidiasis (mostly non-candidemia) [6,53]. In the present study, fluconazole resistance was detected in 6% (2/34) of *C. albicans* isolates from HVSs, which may represent a relatively lower rate. Nevertheless, fluconazole resistance rates among blood isolates were relatively high in *C. albicans* (25%, 1/4) and NAC (34%, 12/35). Because of the low number of *C. albicans* blood isolates in this study, the prevalence of fluconazole resistance in candidemia remains to be determined.

Sequence analyses of the *ERG11* genes of azole-resistant *C. albicans* isolates from blood and HVSs revealed the presence of three amino acid substitutions, among which A114V and F145L in both isolates were located in hotspot I [54] and were associated with increased MIC to fluconazole [32,55]. Although the L276V in hotspot II that was detected within an HVS isolate was a novel mutation, the reported mutations in this region were less frequent, and their contribution to fluconazole resistance was not evident [55,56,57]. Nevertheless, the present analysis of *ERG11* is still preliminary; thus, further, more robust study is necessary to determine the development of *ERG11* mutations in fluconazole-resistant *Candida* in Bangladesh.

The present study is the first characterization of *Candida* spp., along with clinical characteristics/risk factors of candidiasis, in Bangladesh. In this study, sepsis, chemotherapy, and total parenteral nutrition were found to be risk factors, as has been previously described for invasive candidiasis [17,52]. The prolonged use of broad-spectrum antibiotics was found in all of the candidemia cases, along with high frequencies of obstetric or neonatal complications with candidemia, which are considered to be more related to medical status in developing countries. As indicated previously [58], the use of the oral contraceptive pill was the most common risk factor for vulvovaginal candidiasis. These findings, as well as the antifungal susceptibility information within the present study, may contribute to treating and preventing candidiasis in Bangladesh. Further epidemiological study may be necessary to determine the prevalence and trends of *Candida* species and monitor their antifungal resistance, especially for *C. auris*, *K. ohmeri*, and some NAC species.

## Figures and Tables

**Table 1 tropicalmed-07-00211-t001:** Incidence of *Candida* species from three clinical specimens (n = 109).

*Candida* Species	Number of Isolates (% in All the Isolates)	Number of Isolates (% in Specimens)		
		HVS	Blood	Aural Swab
*C. albicans*	40 (36.7)	34 (65.4)	4 (10.3)	2 (11.1)
*C. parapsilosis*	25 (22.9)	4 (7.7)	16 (41)	5 (27.8)
*C. ciferrii*	12 (11)	1 (1.9)	9 (23)	2 (11.1)
*C. tropicalis*	10 (9.2)	5 (9.6)	1 (2.6)	4 (22.2)
*C. glabrata*	6 (5.5)	5 (9.6)	1 (2.6)	0
*C. rugosa*	4 (3.7)	0	4 (10.3)	0
*C. famata*	3 (2.8)	0	0	3 (16.7)
*C. auris*	3 (2.8)	0	3 (7.7)	0
*K. ohmeri*	2 (1.8)	0	0	2 (11.1)
*C. dubliniensis*	1 (0.9)	1 (1.9)	0	0
*C. krusei*	1 (0.9)	1 (1.9)	0	0
*C. kefyr*	1 (0.9)	1 (1.9)	0	0
*C. lusitaniae*	1 (0.9)	0	1 (2.6)	0
Total	109 (100)	52 (100)	39 (100)	18 (100)

**Table 2 tropicalmed-07-00211-t002:** Susceptibility to fluconazole among *C. albicans* and NAC.

*Candida* Species	Number of Isolates	Numbr of Isolates Showing Susceptibility * (% in *C. albicans*/NAC)		
		S	SDD	R
*C. albicans*	40	26 (65)	10 (25)	4 (10)
*C. parapsilosis*	25	17	5	3
*C. tropicalis*	10	5	4	1
*C. glabrata*	6	0	4	2
*C. ciferrii*	12	2	2	8
*C. rugosa*	4	0	4	0
*C. auris*	3	0	0	3
*C. famata*	3	3	0	0
*C. dubliniensis*	1	1	0	0
*C. krusei*	1	0	0	1
*C. kefyr*	1	0	0	1
*C. lusitaniae*	1	1	0	0
*K. ohmeri*	2	0	0	2
NAC total	69	30 (43)	19 (28)	20 (29) **

* S, susceptible; SDD, susceptible dose dependent; R, resistant. ** Significantly higher rate (*p* < 0.05) than *C. albicans*.

**Table 3 tropicalmed-07-00211-t003:** Susceptibility to antifungals among *Candida* isolates from different specimens.

Specimen	Candida species	Number of Isolates	Number of Isolates (%) Showing Susceptibility to Antifungals									
			Fluconazole			Itraconazole			Voriconazole		Amphotericin B	
			S	SDD	R	S	SDD	R	S	R	S	R
Blood	*C. albicans*	4	3	0	1	4	0	0	4	0	4	0
	*C. parapsilosis*	16	13	0	3	9	4	3	16	0	16	0
	*C. ciferrii*	9	3	0	6	3	0	6	8	1	4	5
	*C. rugosa*	4	4	0	0	4	0	0	4	0	4	0
	*C. auris*	3	0	0	3	0	0	3	2	1	0	3
	*C. glabrata*	1	1	0	0	1	0	0	1	0	1	0
	*C. lusitaniae*	1	1	0	0	1	0	0	1	0	0	1
	*C. tropicalis*	1	1	0	0	1	0	0	1	0	1	0
	Total	39	26 (67%)	0	13 (33%)	23 (59%)	4 (10%)	12 (31%)	37 (95%)	2 (5%)	30 (77%)	9 (23%)
HVS	*C. albicans*	34	31	0	3	23	9	2	29	5	25	9
	*C. glabrata*	5	2	1	2	2	1	2	5	0	3	2
	*C. tropicalis*	5	4	1	0	4	1	0	4	1	4	1
	*C. parapsilosis*	4	4	0	0	4	0	0	4	0	3	1
	*C. ciferrii*	1	0	0	1	0	0	1	0	1	0	1
	*C. dubliniensis*	1	1	0	0	1	0	0	1	0	1	0
	*C. krusei*	1	0	1	0	0	1	0	1	0	0	1
	*C. kefyr*	1	1	0	0	0	1	0	1	0	0	1
	Total	52	43 (82%)	4 (8%)	5 (10%)	34 (65%)	13 (25%)	5 (10%)	45 (87%)	7 (13%)	36 (69%)	16 (31%)
			Fluconazole			Itraconazole			Nystatin		Clotrimazole	
			S	SDD	R	S	SDD	R	S	R	S	R
Aural Swab	*C. albicans*	2	2	0	0	1	1	0	2	0	2	0
	*C. parapsilosis*	5	5	0	0	5	0	0	3	2	3	2
	*C. tropicalis*	4	4	0	0	2	1	1	3	1	2	2
	*C. famata*	3	3	0	0	3	0	0	2	1	2	1
	*C. ciferrii*	2	1	0	1	1	0	1	2	0	1	1
	*K. ohmeri*	2	0	2	0	0	2	0	2	0	0	2
	Total	18	15 (83%)	2 (11%)	1 (6%)	12 (67%)	4 (22%)	2 (11%)	14 (78%)	4 (22%)	10 (56%)	8 (44%)

**Table 4 tropicalmed-07-00211-t004:** Risk factors found among patients with individual candidal infections.

Infection Type/Patient Group	Risk Factors	N (%)
Candidemia, neonates (n = 33)	Prolonged broad spectrum antibiotic	33 (100) *
	Total parenteral nutrition	22 (67) *
	Lower uterine cesarean section	20 (61) *
	Low birth weight	17 (52) *
	Nasogastric feeding	17 (52) *
	O_2_ therapy	17 (52) *
	Preterm birth	17 (52) *
	Neonatal Jaundice	14 (42) *
	History of septicemia	12 (36) *
	Prolonged rupture of membrane	6 (18) *
	Gestational diabetes mellitus	2 (6)
	Meconium stained liquor	2 (6)
Candidemia, pediatric and elderly (n = 6)	Prolonged broad spectrum antibiotic	6 (100) *
	Sepsis	4 (67) *
	History of taking chemotherapy	3 (50) *
	Malignancy	3 (50) *
	ICU stay (>14 days)	2 (33) *
Vulvovaginal candidiasis (n = 52)	Oral contraceptive pill	28 (54) *
	Recent broad spectrum antibiotic	20 (39) *
	Diabetes mellitus	16 (31) *
	Pregnancy	10 (19) *
Otomycosis (n = 18)	Habitual cleaning	12 (67) *
	Antibiotic or steroid drop use	10 (56) *
	Oil instillation in ear	8 (44) *
	Diabetes mellitus	6 (33) *
	Trauma	2 (11)

* Significantly higher rate (*p* < 0.01) than other infection type/patient groups

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
