# Peer review of "Prevalence and Antifungal Susceptibility of Clinically Relevant Candida Species, Identification of Candida auris and Kodamaea ohmeri in Bangladesh"

_tropicalmed, 2022, doi:10.3390/tropicalmed7090211_

Round 1
Reviewer 1 Report
Dear authors,
I read your manuscript concerning the prevalence of individual Candida species and their susceptibility to antifungal drugs for clinical isolates in a tertiary care hospital in Bangladesh. The topic is actual and provides a real-life experience, both for clinicians and microbiologists.
1) Line 46-47, to improve and underline the importance of new at-risk populations, read and cite:
- Campione E, Cosio T, Lanna C, et al. Predictive role of vitamin A serum concentration in psoriatic patients treated with IL-17 inhibitors to prevent skin and systemic fungal infections. J Pharmacol Sci. 2020;144(1):52-56. doi:10.1016/j.jphs.2020.06.003
2) Lines 66-68 should be moved to the results section, not in the introduction.
3) Line 103, correct reference.
4) Institutional Review Board Statement should be cited in the material and method section.
5) MIC, full name, please
6) References 28 doesn’t match with CLSI but with Indian guidelines. Check.
7) Line 124, version is missing.
8) mL, please.
9) Table 4 is inadequate. You must use the statistical association tests.
10) Line 219, otomycosis. Correct
Author Response
I read your manuscript concerning the prevalence of individual Candida species and their susceptibility to antifungal drugs for clinical isolates in a tertiary care hospital in Bangladesh. The topic is actual and provides a real-life experience, both for clinicians and microbiologists.
1) Line 46-47, to improve and underline the importance of new at-risk populations, read and cite: - Campione E, Cosio T, Lanna C, et al. Predictive role of vitamin A serum concentration in psoriatic patients treated with IL-17 inhibitors to prevent skin and systemic fungal infections. J Pharmacol Sci. 2020;144(1):52-56. doi:10.1016/j.jphs.2020.06.003
Response: Thank you for your comment and suggestion. In the revised manuscript, above point was added, citing the reference (J Pharmacol Sci. 2020;144(1):52-56). (Line 48-49)
2) Lines 66-68 should be moved to the results section, not in the introduction.
Response: According to the comment, the above portion was deleted.
3) Line 103, correct reference.
Response: This reference was corrected.
4) Institutional Review Board Statement should be cited in the material and method section.
Response: According to the suggestion, IRB statement was moved to Methods section. (line 87-89).
5) MIC, full name, please
Response: Full name of MIC was added. (line 130)
6) References 28 doesn’t match with CLSI but with Indian guidelines. Check.
Response: Thank you for the comment. I am sorry to cause misunderstanding to readers. In this sentence, we described how we judged susceptibility to fluconazole for NAC species which were not defined by CLSI, and the reference is correct. In the revised manuscript, Method section 2.3 was totally revised to describe susceptibility testing more clearly, so that readers understand well. The indicated sentence was rephrased as “For NAC species of which the standards of fluconazole susceptibility are not defined by CLSI, the criterion of C. albicans was applied as described previously [31].”.
7) Line 124, version is missing.
Response : Thank you for the comment. This software, Clustal Omega is available online. Though we checked carefully the website of Clustal Omega, there was no description of version. Therefore, version is not added to manuscript.
8) mL, please.
Response : Thank you, we corrected it.
9) Table 4 is inadequate. You must use the statistical association tests.
Response: Thank you for the comment. In the revised version, risk factors were statistically compared among infection types and the reltant information were added to Table 4. Significantly higher rates were marked with asterisks in Table 4, which is explained in footnote. Method of statistical analysis had been written in Method section (section 2.5).
10) Line 219, otomycosis. Correct
Response: This word was corrected.
Reviewer 2 Report
These are my suggestions for Authors:
- expand the Introduction part with references to geographical dependence of Candida prevalence (adding one or two paragraphs)
- also, at the end of the Introduction part please, add one paragraph about the novelty and advances of the following study.
- avoid using "we" and "our" throughout the text, all sentences must be in a passive form
- for all used materials provide the full information about manufacturer, including city and country
- add why you used nutrient medium with chloramphenicol in it
- In line 103 a reference is left behind in plain parenthesis
- 2.3. Subsection - provide a better explanation of the method
Author Response
1) expand the Introduction part with references to geographical dependence of Candida prevalence (adding one or two paragraphs)
Response: Thank you for the suggestion. In the revised manuscript, geographical distribution of Candida prevalence was added in a paragraph with two references. (line 57-67)
2) also, at the end of the Introduction part please, add one paragraph about the novelty and advances of the following study.
Response: Thank you for the suggestion. We added the novelty and advances of the following study in the end part of Introduction.
3) avoid using "we" and "our" throughout the text, all sentences must be in a passive form
Response: We corrected above points throughout the manuscript
4) for all used materials provide the full information about manufacturer, including city and country
Response: Thank you. For materials, full information indicate above were added.
5) add why you used nutrient medium with chloramphenicol in it
Response: SDA media with chloramphenicol was used to inhibit growth of bacteria. This was added in Methods section. (Section 2.2, first para.)
6) In line 103 a reference is left behind in plain parenthesis
Response: This was corrected.
7) 2.3. Subsection - provide a better explanation of the method
Response: This section was totally revised for readers to understand better and exactly.